# Comparative analysis of low-Earth orbit (TROPOMI) and geostationary (GeoCARB, GEO-CAPE) satellite instruments for constraining methane emissions on fine regional scales: application to the Southeast US

Jian-Xiong Sheng[1], Daniel J. Jacob[1], Joannes D. Maasakkers[1], Yuzhong Zhang[1,2], and Melissa P. Sulprizio[1]

[1]School of Engineering and Applied Sciences, Harvard University, Cambridge, MA, USA
[2]Environmental Defense Fund, Washington DC, USA

**Abstract.** We conduct observing system simulation experiments (OSSEs) to compare the ability of future satellite measurements of atmospheric methane columns (TROPOMI, GeoCARB, GEO-CAPE) for constraining methane emissions down to the 25 km scale through inverse analyses. The OSSE uses the GEOS-Chem chemical transport model ($0.25° \times 0.3125°$ grid resolution) in a 1-week simulation for the Southeast US with 216 emission elements to be optimized through inversion of synthetic satellite observations. Clouds contaminate 73-91% of the viewing scenes depending on pixel size. Comparison of GEOS-Chem to TCCON surface-based methane column observations indicates a model transport error standard deviation of 12 ppb, larger than the instrument errors when aggregated on the 25 km model grid scale, and with a temporal error correlation of 6 hours. We find that TROPOMI ($7 \times 7$ km$^2$ pixels, daily return time) can provide a coarse regional optimization of methane emissions, comparable to results from an aircraft campaign (SEAC[4]RS), and is highly sensitive to cloud cover. The geostationary instruments can do much better and are less sensitive to cloud cover, reflecting both their finer pixel resolution and more frequent observations. The information content from GeoCARB toward constraining methane emissions increases by 20-25% for each doubling of the GeoCARB measurement frequency. Temporal error correlation in the transport model moderates but does not cancel the benefit of more frequent measurements for geostationary instruments. We find that GeoCARB observing twice a day would provide 70% of the information from the nominal GEO-CAPE mission pre-formulated by NASA in response to the Decadal Survey of the US National Research Council.

## 1 Introduction

Methane is the second most important anthropogenic greenhouse gas after $CO_2$ (Myhre et al., 2013), and plays a key role in tropospheric and stratospheric chemistry (Thompson, 1992; West and Fiore, 2005; Solomon et al., 2010). The contributions from different source sectors and regions to the atmospheric methane budget remain highly uncertain (Kirschke et al., 2013; Saunois et al., 2016; Turner et al., 2017). Satellite observations of atmospheric methane columns in the shortwave infrared (SWIR) are a promising resource for quantifying emissions through inverse analyses (Jacob et al., 2016; Houweling et al., 2017) but can be limited by instrument precision, sampling frequency, pixel resolution, cloud cover, and model transport error.

Here we apply an Observing System Simulation Experiment (OSSE) for the Southeast US to compare the ability of new satellite instruments to characterize methane emissions down to the 25-km scale, using as reference results from the recent SEAC[4]RS aircraft campaign in the region (Sheng et al., 2018).

SWIR methane observations from space have so far been mainly from the SCIAMACHY instrument (2003-2013; Frankenberg et al., 2006) and the GOSAT instrument (2009-2016; Kuze et al., 2009, 2016). These data have proven useful for optimizing methane emissions on regional scales down to $\sim$100 km when averaged over several years (Bergamaschi et al., 2013; Fraser et al., 2013; Monteil et al., 2013; Wecht et al., 2014b; Turner et al., 2015; Alexe et al., 2015; Feng et al., 2017), but they are too sparse to constrain methane emissions on finer spatial or temporal scales. Our ability to observe methane from space should be considerably improved with the recent launch (October 2017) of the SWIR TROPOMI instrument, providing daily global coverage with 0.6% precision and $7 \times 7$ km$^2$ nadir resolution (Butz et al., 2012; Hu et al., 2018). The GeoCARB geostationary mission to be launched in early 2020s plans to observe methane columns over North and South America with 0.6% precision and $3 \times 3$ km$^2$ resolution (Polonsky et al., 2014; O'Brien et al., 2016). The observing frequency of GeoCARB is not finalized yet and could be 1-4 times per day. Other geostationary instruments still at the proposal stage offer improved combinations of pixel size, precision, and observing frequency, including GEO-CAPE (Fishman et al., 2012), GeoFTS (Xi et al., 2015), G3E (Butz et al., 2015), and CHRONOS (Edwards et al., 2018). GEO-CAPE has been pre-formulated by NASA as a recommended mission from the US National Research Council (2007) Decadal Survey on Earth Science and Applications from Space.

OSSEs are standard approaches to assess the utility of future satellite instruments to deliver on a specific objective, here the mapping of methane emissions. OSSEs at 50 km spatial resolution have been conducted to evaluate the potential of future satellite observations for quantifying methane emissions over California (Wecht et al., 2014a) and North America (Bousserez et al., 2016). Bousserez et al. (2016) assessed the benefit of geostationary multi-spectral (SWIR + thermal infrared) measurements. Turner et al. (2018) conducted a kilometer-resolution OSSE to explore the potential of different satellite observing configurations to resolve the distribution of methane emissions on the scale of an oil/gas field, and Cusworth et al. (2018) extended that work to examine the ability of the satellites to detect anomalous high-mode point source emitters.

Here we conduct a comparative analysis of TROPOMI, GeoCARB, and GEO-CAPE for constraining the spatial distribution of methane emissions at a fine regional scale (25 km), and we investigate more generally how the information content from different satellite observing configurations depends on pixel size, observing frequency, and cloud contamination. Of particular interest is to define observing frequency requirements for GeoCARB to resolve regional-scale methane sources. We focus on the Southeast US, which accounts for about 50% of US methane emissions including mixed contributions from wetlands, fossil fuels, agriculture, and waste (Maasakkers et al., 2016; Bloom et al., 2017). Sheng et al. (2018) previously used boundary layer methane observations from the NASA SEAC[4]RS aircraft campaign (Toon et al., 2016) in August-September 2013 to optimize methane emissions over the Southeast US. This offers an opportunity to directly compare the observing power of satellite instruments to that from a dedicated aircraft campaign.

## 2 Observing system simulation experiments

Our OSSE framework is shown in Figure 1. We build on the previous work of Sheng et al. (2018), who conducted a Bayesian inverse analysis of the SEAC$^4$RS aircraft observations with the GEOS-Chem chemical transport model (CTM) at $0.25° \times 0.3125°$ resolution. They used the SEAC$^4$RS data together with prior estimates and error statistics from the gridded EPA inventory of Maasakkers et al. (2016) and the WetCHARTs extended ensemble wetland inventory of Bloom et al. (2017), to optimize the spatial distribution of methane emissions in the Southeast US for August-September 2013. We follow the same analytical inversion framework as Sheng et al. (2018) for our OSSE. We first simulate a methane column concentration field using the GEOS-Chem CTM with prior emission estimates (base simulation). We then sample this field following the specifications of the different satellite instruments (Table 1), accounting for instrument random noise and cloud contamination (discussed below).

For TROPOMI we assume a $7 \times 7$ km$^2$ pixel size, which is the design nadir value (Butz et al., 2012); actual pixel sizes grow toward the outer parts of the cross-track swath. On the other hand, there are plans to deliver TROPOMI data at finer $5.5 \times 7$ km$^2$ pixel resolution (Ilse Aben, SRON, personal communication). The $3 \times 3$ and $4 \times 4$ km$^2$ pixel resolutions assumed for GeoCARB and GEO-CAPE are generic values for the contiguous US in the current designs. Randomness in the noise of synthetic observations is a standard OSSE assumption (e.g., Wecht et al., 2014a; Bousserez et al., 2016) but may overestimate the information in the observations if some of the actual noise is systematic (Bousquet et al., 2018).

The sampled synthetic observations define the observation vector $\mathbf{y}$ for the inversion. The sensitivity of these observations to the distribution of methane emissions over the domain (arranged as a state vector $\mathbf{x}$) is defined by the Jacobian matrix $\mathbf{K} = \partial\mathbf{y}/\partial\mathbf{x}$, where the $i$th column of $\mathbf{K}$ ($\partial\mathbf{y}/\partial x_i$) is constructed from GEOS-Chem by perturbing individual state vector elements $x_i$ to compute the resulting perturbation $\Delta\mathbf{y}$ (relative to the base simulation). We then use this Jacobian matrix together with prior and observational error statistics (error covariance matrices $\mathbf{S_A}$ and $\mathbf{S_O}$) to quantify the information content of observations toward constraining emissions. All observations use a mean SWIR averaging kernel from GOSAT with uniform near-unit sensitivity in the troposphere (Worden et al., 2015). The OSSE is conducted for the one-week period of August 8-14, 2013. Although this observation period is relatively short (limited by the OSSE cost of computing the Jacobian matrix), it provides useful comparison of the different satellite observing configurations and their sensitivities to measurement frequency and cloud cover. A longer observing period would provide more information.

The state vector $\mathbf{x}$ of emissions, representing the spatial distribution of emissions to be resolved by the inversion, is the same as in Sheng et al. (2018). It includes 216 Gaussian mixture model (GMM) elements, where each element is a Gaussian mode with radial basis functions (RBFs) applied to the $0.25° \times 0.3125°$ grid (Turner and Jacob, 2015). The modes are selected on the basis of criteria including spatial proximity and source type patterns as in Turner and Jacob (2015). The optimization is for the amplitudes of the 216 Gaussian modes, and the corresponding solution on the $0.25° \times 0.3125°$ grid is obtained from the RBF weights. In this manner, each $0.25° \times 0.3125°$ grid cell is individually optimized as a linear combination of Gaussian modes with RBFs. Figure 2 shows the resulting approximate clustering as the grid cells whose largest RBF weights are for common Gaussian modes. We choose to optimize 216 elements as representing the extent of information on emissions that we may hope

to achieve with 1-week observations. The use of the GMM with RBFs allows us to resolve localized dominant sources (such as oil/gas or coal mines) at high resolution while degrading resolution in areas of weak or broadly distributed sources. The GMM also reduces errors in aggregation of the state vector as compared to a simple grid coarsening method (e.g., 216 elements at $1° \times 1.25°$ resolution), which would mix neighboring source types and induce larger aggregation error.

The analytical solution to the Bayesian inversion problem includes full characterization of the information content from the observations towards quantifying the state vector of emissions, as computed by the Degrees of Freedom For Signal (DOFS; Rodgers, 2000). Combining the Jacobian matrix $\mathbf{K}$ constructed from GEOS-Chem together with the prior error covariance matrix $\mathbf{S_A}$ and the observation error covariance matrix $\mathbf{S_O}$, we compute the averaging kernel matrix $\mathbf{A} = \partial \hat{\mathbf{x}}/\partial \mathbf{x}$ that represents the sensitivity of the optimization ($\hat{\mathbf{x}}$) to the true state ($\mathbf{x}$):

$$\mathbf{A} = \mathbf{S_A}\mathbf{K}^T(\mathbf{K}\mathbf{S_A}\mathbf{K}^T + \mathbf{S_O})^{-1}\,\mathbf{K} = \mathbf{I_n} - \hat{\mathbf{S}}\mathbf{S_A}, \qquad (1)$$

where $\mathbf{I_n}$ is the identity matrix of dimension $n$ (=216) and $\hat{\mathbf{S}}$ is the posterior error covariance matrix. The DOFS is then the trace of the averaging kernel matrix:

$$\text{DOFS} = \text{tr}(\mathbf{A}) = \text{tr}(\mathbf{I} - \hat{\mathbf{S}}\mathbf{S_A}). \qquad (2)$$

The DOFS represents the number of pieces of information provided by the observing system for quantifying the state vector.
As seen from Equation (2), the DOFS is related to the relative reduction in error variance that would be obtained from the ratios of the diagonal elements of $\hat{\mathbf{S}}$ and $\mathbf{S_A}$ . It provides however a more complete characterization of information content by accounting for error covariances. DOFS = 216 would represent perfect constraints on our state vector. The SEAC[4]RS aircraft inversion of Sheng et al. (2018) achieved DOFS = 10.

The prior error covariance matrix $\mathbf{S_A}$ for our problem is taken from the emission inventory error estimates of Maasakkers
et al. (2016) for anthropogenic sources and Bloom et al. (2017) for wetlands, as described by Sheng et al. (2018). The observational error covariance matrix $\mathbf{S_O}$ is specific to the observing configuration, and includes contributions from model transport error in simulating the observations as well as the instrument errors given in Table 1.

We estimate the model transport error variance by the residual error method (Heald et al., 2004) applied to the GEOS-Chem simulation with prior emissions of hourly observed Total Carbon Column Observing Network (TCCON) methane columns in
Lamont, Oklahoma for August-September 2013 (Wunch et al., 2011; Wennberg et al., 2017). In that method, the mean bias in the model compared to the observations is attributed to error in the prior emissions (to be corrected in the inversion) and the residual characterizes the observation error including contributions from both model transport error and instrument error. In our case, the TCCON measurements are highly precise (precision is <4 ppb), so that the residual characterizes the model transport error. The residual error distribution is shown in Figure 3 and features an error standard deviation of 12 ppb. This
error standard deviation is consistent with previous GEOS-Chem transport error estimates by the residual error method using GOSAT observations from Wecht et al. (2014a) for California and Turner et al. (2015) for North America. We assume therefore that it applies over our whole domain.

Temporal correlation in the model transport error may limit the benefit of high-frequency observations, because repeated observations of the same scene may produce the same model-observation differences. Here we estimate this error correlation

from the autocorrelation vs. time lag of the difference between GEOS-Chem and TCCON observations. Results in Figure 3 (right panel) show an exponential fit function with error correlation time scale of 6 hours which we apply as off-diagonal elements in the observational error covariance matrices for the different satellite observing configurations. The increase of the autocorrelation coefficients around 12 hours is possibly due to fewer observations (TCCON observations are only available in the daytime) or neglecting to apply solar-zenith-angle dependent averaging kernels in the modeled column methane, but it does not significantly affect the exponential fit. Figure 4 is the persistence (e-folding) time scale for cloud cover, which affects the extent to which the temporal error correlation limits the information content of high-frequency observations; this will be discussed in the next Section.

The instrument error for individual observations is given by the precision values in Table 1, taken from the original references. The observations are averaged over $0.25° \times 0.3125°$ GEOS-Chem grid cells for the purpose of the inversion, and the instrument error standard deviation is decreased by the square root of the number of successful retrievals averaged over each grid cell for individual retrieval time.

Any cloud contamination within an observation pixel will cause an unsuccessful SWIR retrieval for methane (Butz et al., 2012). Remer et al. (2012) used high-resolution cloud data (0.5-1.0 km) over the US for different regions and seasons to infer probabilities for satellites to view clear-sky as a function of pixel size. They focused on aerosol retrievals and here we use their same statistics for methane retrievals. For the Southeast US in summer with an average cloud fraction of 0.7, we find that cloud contamination would invalidate 91% of retrievals for TROPOMI ($7\times7$ km$^2$ pixels), 73% for GeoCARB ($3\times3$ km$^2$ pixels), and 79% for GEO-CAPE ($4\times4$ km$^2$ pixels). Slant light paths and 3-D cloud scattering would further decrease the frequency of successful retrievals. Our OSSE retrieval failure rate of 91% for TROPOMI in the Southeast US is similar to the global mean failure rate of 92% for the GOSAT ($10 \times 10$ km$^2$) full-physics retrieval (Parker et al., 2011; Schepers et al., 2012). Sensitivity to retrieval success rate will be discussed in the next section through modifications of cloud cover.

Our removal of cloudy observations uses three-hour $0.25° \times 0.3125°$ fractional cloud cover information in the GEOS-FP meteorological data driving GEOS-Chem (Lucchesi, 2013), and then scales the removal rates regionally to match the cloud contamination rates in Table 1. Although the satellite data loss from cloud cover is severe, the relatively coarse $0.25° \times 0.3125°$ resolution of our inversion allows aggregation of data from a large number of observation pixels for comparison to the model. This does not help when there is solid cloud cover on the 25 km scale in the GEOS-FP data (as in the white areas for the GeoCARB pseudo-observations in Figure 1) but it helps for fractional cloud cover. The median number of aggregated successful pixel retrievals for a given $0.25° \times 0.3125°$ grid cell at a given observation time is 3, 30, and 15 for TROPOMI, GeoCARB, and GEO-CAPE, respectively. Thus the median instrument error standard deviation on the $0.25° \times 0.3125°$ grid scale over our inversion domain is 6 ppb for TROPOMI, and 2-4 ppb for the geostationary instruments. This is smaller than the 12 ppb model transport error standard deviation (Figure 3), so that most of the observational error is contributed by model transport. This is an important result as it implies that inversion results are relatively insensitive to instrument precision at the 25 km scale. Turner et al. (2018) found much more sensitivity to satellite instrument precision when attempting to optimize emissions at kilometer scales.

## 3   Results and discussion

The information content from different satellite observing configurations is diagnosed by the DOFs, as described in the Methods section, representing the number of pieces of information on emissions that can be retrieved by inversion of synthetic observations. Figure 5 shows a contour plot of the DOFS as a function of observing frequency and pixel resolution, assuming a fixed instrument precision of $0.6\%$. As discussed in the previous Section, results are relatively insensitive to instrument precision since most of the observational error is contributed by model transport. The DOFS increase as measurement frequency increases (more independent observations) and as pixel size decreases (more observations aggregated in a $0.25° \times 0.3125°$ grid cell). The benefit of increasing measurement frequency eventually weakens at high values because of temporal correlation in the GEOS-Chem model transport error. The benefit of increasing pixel resolution also weakens below 4 km because the inversion does not try to resolve emissions to resolution finer than $0.25° \times 0.3125°$. Even so, the maximum DOFS of 70 in Figure 5 that could be achieved by a very high-resolution system (1 km pixel size and hourly observations) is much less than the ideal value of 216 representing full characterization of the emission field. This is because we only use one week of observations.

DOFS for TROPOMI, GeoCARB (1-4 measurements per day) and GEO-CAPE are indicated on the contour map. The TROPOMI inversion has 26 DOFS, higher than the SEAC[4]RS aircraft campaign (DOFS = 10; Sheng et al., 2018). The geostationary GeoCARB and GEO-CAPE observations achieve higher DOFS, reflecting their higher observing frequency and pixel resolution (greater density of observations). The GeoCARB information content increases by about 20% when going from 1 to 2 measurements for day, and another 20% when going from 2 to 4 measurements per day. GEO-CAPE provides higher DOFS than GeoCARB, despite coarser pixels, because it measures hourly. We see from Figure 5 that an instrument measuring hourly with $7 \times 7$ km$^2$ pixels would provide the same information as GeoCARB measuring 4 times per day with $3 \times 3$ km$^2$ pixels, and GeoCARB measuring twice a day would provide about 70% of information content obtained from GEO-CAPE hourly measurements. Again, this result depends on the spatial resolution of the inverse problem (here $\sim$25 km). A focus on resolving emissions on finer scales would place a larger premium on decreasing pixel size.

Figure 6 (left panel) examines further the sensitivity of the DOFS to observing frequency for GeoCARB, and the role of the model transport error correlation in limiting the gains from increasing measurement frequency. Without model transport error correlation the DOFS increase roughly as the square root of the measurement frequency (about 40% for each doubling), as would be expected from the central limit theorem. Temporal error correlation significantly reduces but does not eliminate the gain from increasing observing frequency. Thus we find that the DOFS increase by 20-25% instead of 40% for each doubling of the measurement frequency when temporal error correlation is taken into account. Beyond increasing data density, an advantage of more frequent measurements for a region is to increase the opportunity for observing clear-sky scenes ("cloud clearing"), particularly if clouds are more transient than the 6-hour error correlation time scale (in which case multiple observations over that time scale would increase the chance of obtaining a clear-sky value). Cloud cover in the GEOS-FP meteorological data used to drive GEOS-Chem has a persistence time scale typically longer than 6 hours (Figure 4), which moderates this cloud-clearing benefit of high-frequency observations.

All satellite observing configurations considered in our work have low retrieval success rates because of cloud contamination of individual pixels (Table 1), as determined from the Remer et al. (2012) clear-sky probability statistics for the Southeast US. These statistics are for summer (regional cloud cover of 70%), but Remer et al. (2012) also give statistics for other seasons with regional cloud cover for the Southeast ranging from 55 to 81%. Figure 6 (right panel) shows the effects of these different cloud statistics on the DOFS for the TROPOMI, GeoCARB, and GEO-CAPE configuration. TROPOMI ($7 \times 7$ km$^2$) is strongly sensitive to regional cloud cover because of its coarse pixel size and (to a lesser extent) its infrequent return time. The geostationary systems are far less sensitive to cloudy conditions. The effects of clouds on the information content of TROPOMI is further illustrated in Figure 7 with the averaging kernel sensitivities (diagonal elements of the averaging kernel matrix) relative to clear-sky conditions. The loss of information varies by region depending on the extent of cloud cover.

## 4  Conclusions

We performed observing system simulation experiments (OSSEs) to compare the ability of low-Earth orbit (TROPOMI) and geostationary (GeoCARB, GEO-CAPE) satellite instruments for constraining methane emissions through inverse analyses. The OSSEs use the GEOS-Chem chemical transport model ($0.25° \times 0.3125°$ grid resolution) in a 1-week simulation for the Southeast US with 216 emission state vector elements. The information content from the different satellite instrument configurations towards quantifying the state vector of emissions is computed as the degrees of freedom for signal (DOFS) using a Bayesian analytical inversion framework.

We find that inverse analysis of TROPOMI observations of atmospheric methane columns should provide a successful regional characterization of methane emissions, though with limited spatial resolution. The information content from TROPOMI is strongly dependent on cloud cover, due to limited cloud-clearing capability (coarse pixels, infrequent return time). Geostationary observations can perform much better, with less dependence on cloud cover, due to a combination of finer pixel resolution and more frequent returns. GeoCARB gains 20-25% in information content for each doubling of its measurement frequency from once to eight times per day. GeoCARB measuring twice a day can deliver 70% of information content from the GEO-CAPE configuration (hourly observations). The benefit of increasing observation frequency is moderated by the 6-h temporal error correlation in the transport model.

*Acknowledgements.* This work was funded by the NASA Earth Science Division. We thank Alexander J. Turner for helpful discussion. Yuzhong Zhang's work was partially funded by the Kravis Scientific Research Fund at the Environmental Defense Fund. TCCON data were obtained from the TCCON Data Archive, hosted by CaltechData (http://tccondata.org)

**Table 1.** Specifications of satellite instruments.[a]

| Instrument | Observing frequency [b] | Pixel size ( km$^2$ ) | Precision | Cloud contamination [c] | Reference |
|---|---|---|---|---|---|
| TROPOMI | once a day | 7×7 | 0.6% | 91% | Butz et al. (2012) |
| GeoCARB | 1-4 times a day | 3×3 | 0.6% | 73% | Polonsky et al. (2014); O'Brien et al. (2016) |
| GEO-CAPE | once an hour | 4×4 | 1% | 79% | Fishman et al. (2012) |

[a] All instruments measure atmospheric methane columns with near-uniform sensitivity in the troposphere, specified here with a typical SWIR averaging kernel (Worden et al., 2015).

[b] All observations are daytime only (SWIR solar back-scatter instruments) and limited to the 9:00-16:00 local time (LT) window. TROPOMI observes at 13:00 LT once a day. GeoCARB observes at 13:00 LT (once a day), 11:00 and 13:00 LT (twice a day), or 9:00, 11:00, 13:00, and 15:00 LT (four times a day). GEO-CAPE observes every hour in the 9:00-16:00 LT window (8 times a day).

[c] Percentage of observing scenes with unsuccessful retrievals due to cloud contamination (Remer et al., 2012).

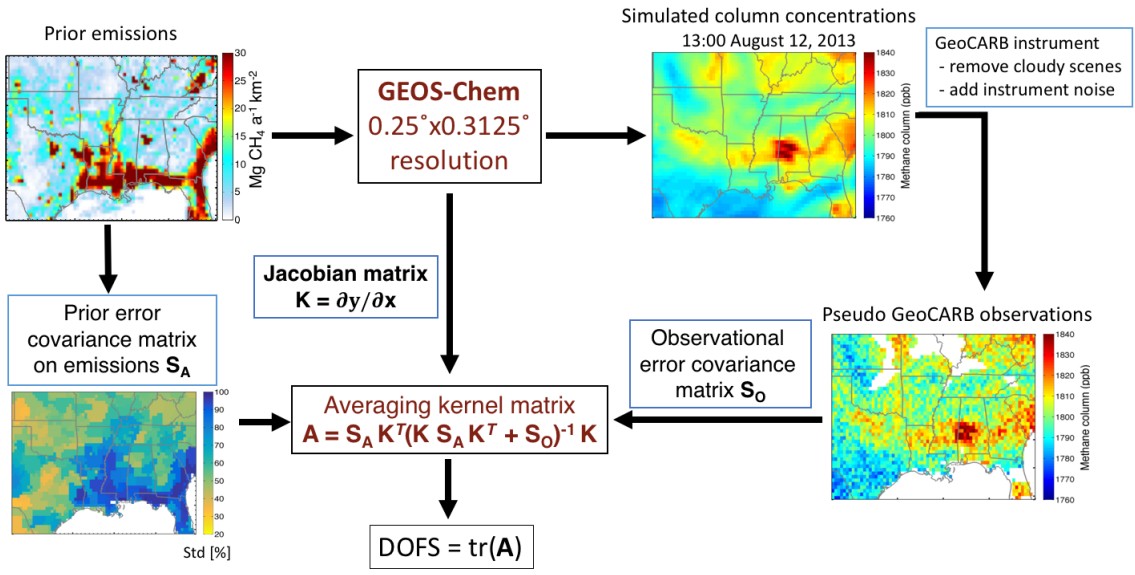

**Figure 1.** Observing System Simulation Experiment (OSSE) framework for the Southeast US to compare the ability of new satellite instruments to constrain methane emissions on the 25 km ($0.25° \times 0.3125°$) scale. GeoCARB is used here as an example. The right panels show illustrative column concentrations and corresponding GeoCARB observations for a particular time. The column concentrations are in unit of dry molar mixing ratio (ppb). White areas indicate full cloud cover or oceans preventing GeoCARB from making any observations on the 25 km scale. The prior error covariance matrix on emissions $S_A$ is assumed diagonal and shown here as the corresponding relative error standard deviations. The Degrees Of Freedom for Signal (DOFS) is the trace of the averaging kernel matrix and measures the information content from the different satellite instruments.

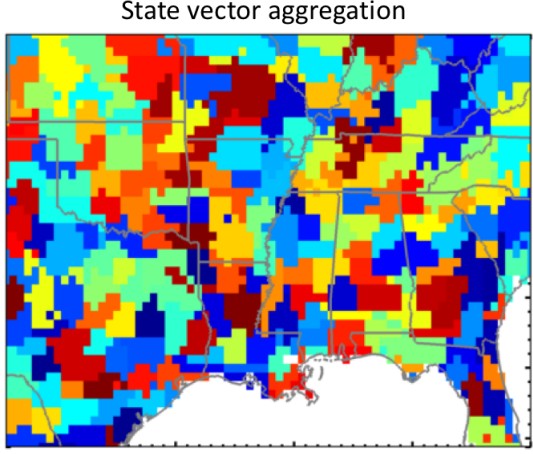

**Figure 2.** Approximate rendition of the reduced-dimension state vector of $n = 216$ elements used to constrain methane emissions in the Southeast US. This reduced-dimension state vector was obtained by projecting the 3456 GEOS-Chem grid cells at $0.25° \times 0.3125°$ resolution onto a Gaussian mixture model (GMM) with radial basis functions (RBFs), as described in the text. The colors group together $0.25° \times 0.3125°$ grid cells with largest RBFs for a given Gaussian mode and have no other significance. This visualization of the state vector as a cluster with hard boundaries is an approximate rendition because each $0.25°.3125°$ grid cell is in fact individually optimized as a superimposition of the 216 Gaussian modes with RBF weights.

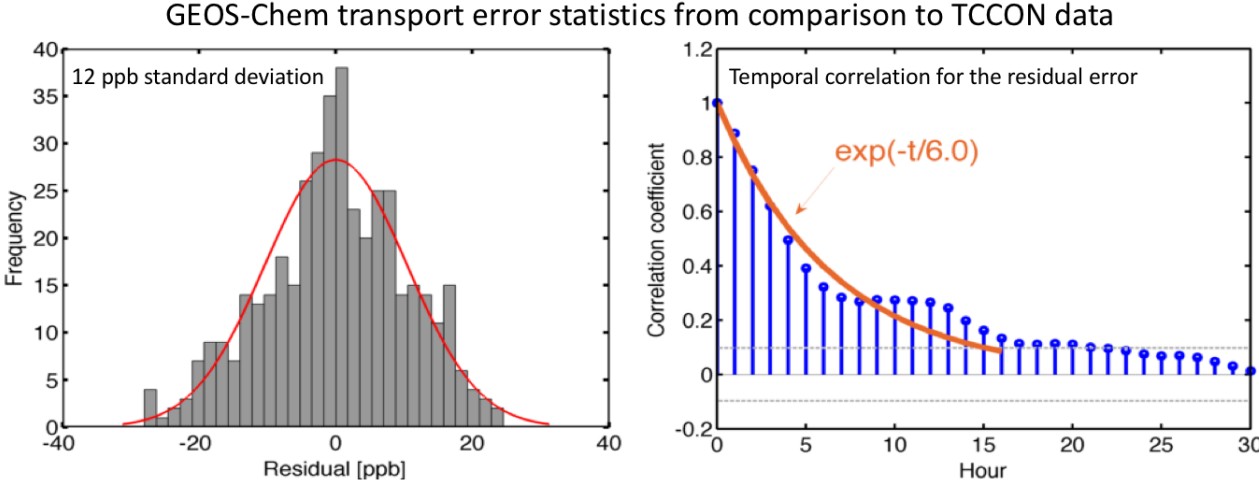

**Figure 3.** GEOS-Chem model transport error statistics derived from the residual error method (Heald et al., 2004) applied to hourly TCCON ground-based observations in Lamont, Oklahoma, in August-September 2013. Residuals are the differences between hourly simulated and observed values after removal of the mean bias. The left panel shows the frequency distribution of residual error (GEOS-Chem minus TCCON) and a Gaussian fit to that distribution with standard deviation 12 ppb. The right panel panel shows autocorrelation coefficients of the residual error plotted against time lag and an exponential fit with a temporal error correlation e-folding scale of 6 hours. Significance levels ($p < 0.05$) are shown as dashed lines. The correlation becomes insignificant past a time lag of 16 hours.

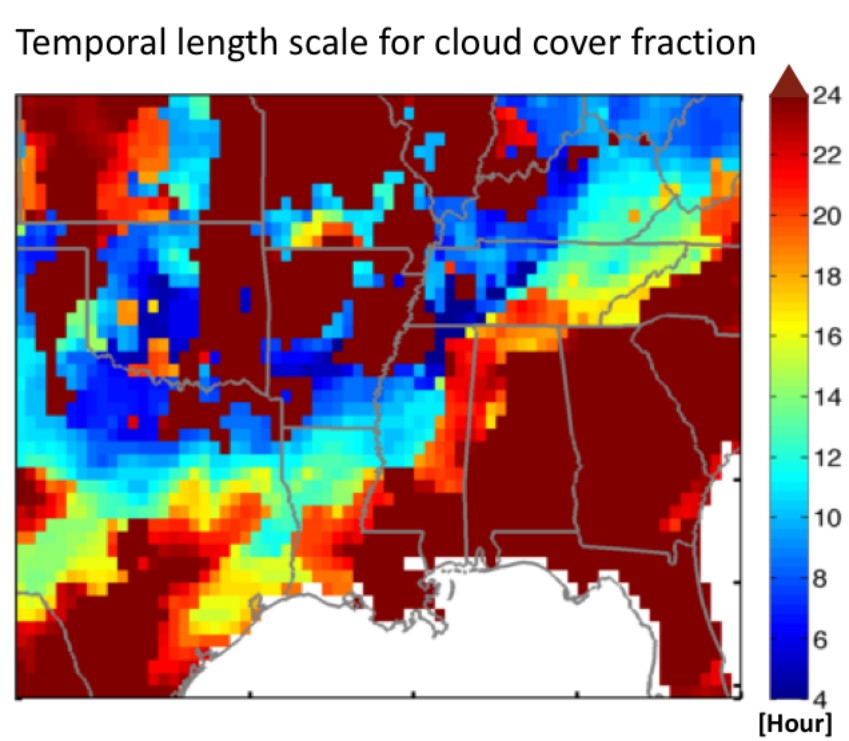

**Figure 4.** Persistence time scale for cloudy conditions in the GEOS-FP assimilated meteorological data for August-September 2013. The persistence time scale is defined as the temporal e-folding correlation time scale for total cloud cover fraction in the 3-hour GEOS-FP data.

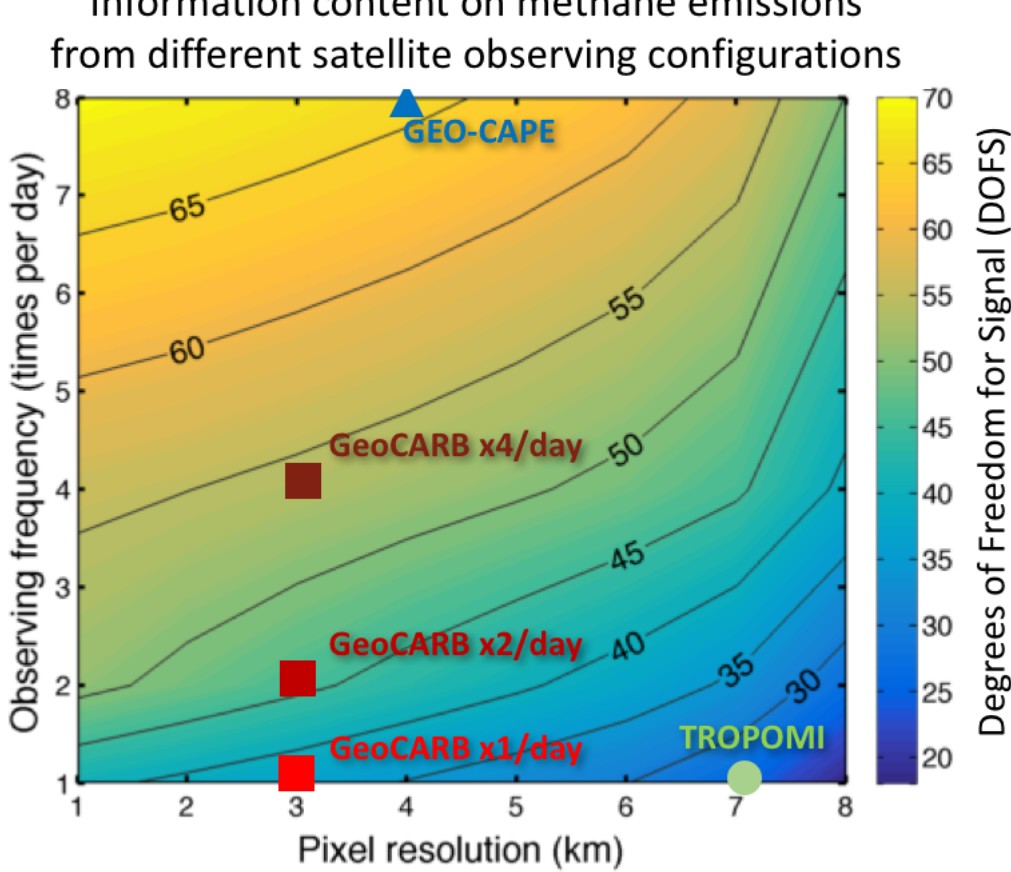

**Figure 5.** Information content of different satellite observing configurations for constraining the distribution of methane emissions in the Southeast US. The figure shows the Degrees of Freedom for Signal (DOFS) for a 1-week observation period aiming to constrain 216 emission elements in the Gaussian mixture model characterizing the distribution of emissions at up to 25 km resolution. The configurations are defined by their observing frequency and pixel resolution. The DOFS for the TROPOMI, GeoCARB (1, 2, and 4 measurements per day), and GEO-CAPE observations are indicated.

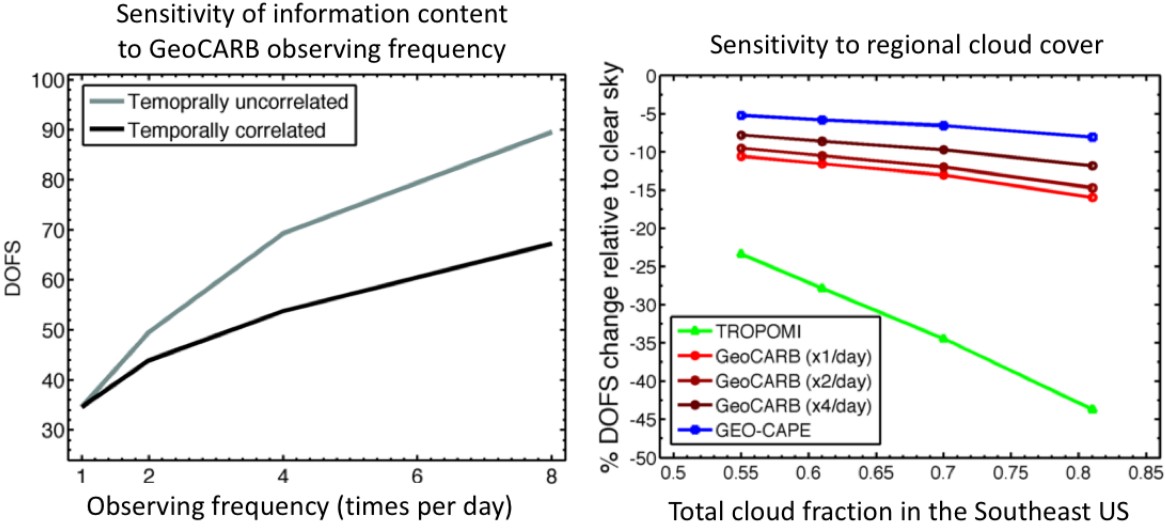

**Figure 6.** Effects of observing frequency and regional cloud cover on the information content (DOFS) from different satellite observing configurations in constraining methane emissions on the 25 km scale.. The left panel shows the sensitivity of the DOFS to observing frequency for the GeoCARB instrument, with and without accounting for temporal correlation in the model transport error (e-folding time scale of 6 hours). The right panel shows the sensitivity of the DOFS to regional cloud fraction, as a percentage decrease relative to clear sky, using the combination of the GEOS-FP cloud cover data and clear-sky probabilities as a function of pixel size (Remer et al., 2012).

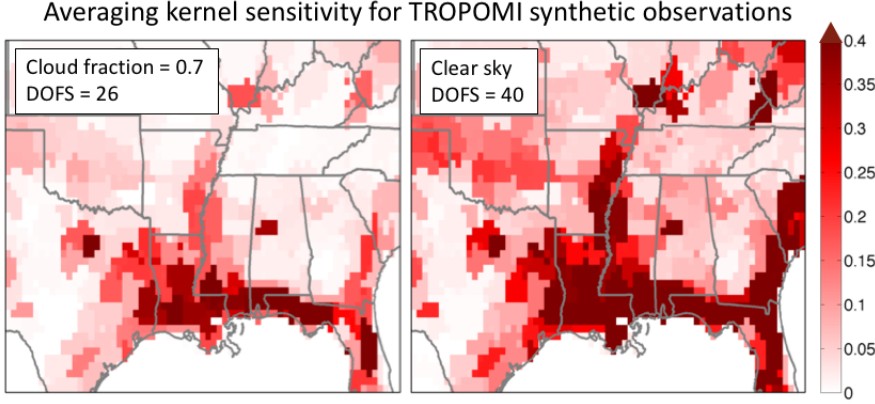

**Figure 7.** Diagonal elements of the averaging kernel matrix from our OSSE using TROPOMI synthetic observations under cloudy (cloud fraction = 0.7; left panel) and clear-sky conditions (right panel), representing the ability of the observations to constrain local emissions (see text). The sum of these values (trace of the average kernel matrix) is the DOFS of the inversions.

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
