# Peer review of "Comparative analysis of low-Earth orbit (TROPOMI) and geostationary (GeoCARB, GEO-CAPE) satellite instruments for constraining methane emissions on fine regional scales: application to the Southeast US"

_Atmospheric Measurement Techniques, 2018_

## Referee Comment (RC1) · Anonymous Referee #1 · 1 Jul 2018

The paper by Sheng et al. examines the information content on methane ($CH_4$) emissions contained in column-average concentration measurements by three satellite configurations. These configurations reflect the TROPOMI mission in low-Earth-orbit and the GeoCarb and GeoCAPE missions in geostationary orbit. The information content is estimated by a Bayesian inversion for simulated measurements above the Southeast US for a week in summer.

The paper is well written and interesting for the readers of Atmospheric Measurement Techniques in particular since the study can serve as reference for how to size future satellite techniques in terms of spatiotemporal resolution. Therefore, I recommend publishing the paper after considering my comments below:

- The paper is a case study for 1 week of $CH_4$ emissions in the Southeast USA. How representative is this case study for the overall challenge of inversely estimating methane emissions on regional scales globally for all seasons? The study would gain scientific mass by extending to other regions and other seasons.

- The Bayesian inversion essentially is controlled by the weighting between the measurement uncertainty and the a priori uncertainty. While the assumed measurement uncertainties are described in quite some detail, the text is sparse for the a priori uncertainties. I recommend elaborating in more detail how large the assumed a priori uncertainties are, e.g. a map would help. Is the uncertainty relative to the a priori fluxes i.e. vanishing a priori fluxes remain zero? Likewise, it would be helpful to illustrate the effect of Gaussian Mixture Model used for spatial binning. This information should be included even if it is redundant with previous publications.

- The performance analysis focusses on the DOF which is a very condensed measure. I would recommend extending the analysis to the a posteriori flux errors (or the error reductions wrt. the a priori). Could it be enlightening to plot the averaging kernel matrix for cloudy and less cloudy conditions to illustrate the effects of clouds on the information content?

- The assumed gorund-pixel sizes (table 1) are valid for the sub-satellite point (to the best of my knowledge). For wide-swath LEO missions such as TROPOMI, ground-pixels grow substantially toward the outer parts of the swath. Likewise for GEO, ground-pixel sizes grow with latitude and longitude away from the sub-satellite point. In that sense, the study is too optimistic with respect to the real satellite performance (cloud contamination, measurement density).

- Figure 2, right panel: The inset is somewhat misleading since intuitively one would expect the inset to show kind of the same quantity as the main figure. But, in fact, it is TCCON-model departures in the main figure and cloud cover in the inset. It took me a while to get it. Consider making it separate figures.

- The most recent publication for TROPOMI $CH_4$ (real data) is Hu et al., https://doi.org/10.1002/2018GL077259, 2018

---

## Referee Comment (RC2) · J. Marshall (Referee) · 15 Aug 2018

This paper presents in a very compact nature a methodology for comparing three different satellite missions working to constrain methane fluxes using Degrees of Freedom for Signal (DOFS) as a metric of the resolvable information content. This methodology is applied in OSSEs looking at the relative performance of TROPOMI, GeoCARB, and GEO-CAPE. The approach is interesting, and provides a slightly different assessment

than the usual reduction of posterior uncertainty. Nonethesless, I have a few concerns regarding some of the assumptions made (particularly with respect to the influence of cloud cover on measurement yield) and the presentation of the results. While the writing is quite clear and free from errors, some additional information is required to help the reader truly understand the approach. (Perhaps the manuscript is a bit too compact?) Even after reading some of the referenced papers in a search for explanation, the interpretation of the results was somewhat difficult. As such, some additional information is requested, as outlined below. If these points are addressed, I would consider the paper suitable for publication in AMT.

In particular, some physical interpretation of the state vector elements developed using the Gaussian Mixture Method (GMM) with Radial Basis Functions (RBFs) would be helpful. The paper in which this method was developed (Turner and Jacob, 2015) is mathematically rather dense, but does provide some information about what these functions look like for California. Having some idea about the relevant processes and the spatial distributions that might be resolved in the study domain used here would be useful. Was temporal aggregation performed as well, over the week, or was a stationary solution assumed? How would this methodology be extended to different time periods or regions? Would the state vector have signficantly more or less elements for other similarly-sized domains?

P3, L7-9: I disagree with the statement that the assumption of randomness in the noise of synthetic measurements does not affect a comparative analysis of different instruments. This is true if the instruments which are being compared are expected to have similarly correlated or uncorrelated errors in their actual measurements, but this may well not be the case. An example of this is active vs. passive sensors, where the former is expected to have considerably less correlation between individual measurements. While such an assumption has often been made in the past, more experience with satellite measurements have proven time and again that systematic (correlated) errors are incredibly important when trying to interpret signals, and they are not identical across instruments. Please discuss explicitly the limitations of this assumption.

P4, L3-10: In the discussion of the model transport error variance, the approach seems valid, but I wonder about its broader applicability over the full domain. Other studies (using in situ data) have shown that the uncertainty dominated by transport errors tends to be proportional to the mean mixing ratios for a given period (see e.g. Jeong et al., JGR, 2013). How representative are the signals at Lamont for the whole domain?

P4, L11-17: In ths discussion of the temporal correlation of the measurements, please explain the increase at around 12 hours seen in the second panel of Figure 2. Does this have something to with the fact that the TCCON measurements are made only during the day, and as such there are fewer samples at around 12 hours? Or is this the result of the well-known smily/frowny tendencies of some TCCON sites at highsolar zenith angles? Or an airmass dependency that is not properly accounted for when comparing the modelled fields to the TCCON data? This peak in correlation at 12 hours could be the result of neglecting to apply the (solar-zenith-angle-dependent) averaging kernels to the modelled fields before performing the comparison. Was this done? Some more detail is needed here.

P4, L22-P5, L7: The discussion of the cloud cover is probably the most critical point in this manuscript, upon which many of the conclusions rest. I expect that the estimation of number of successful retrievals is overestimated for partially cloudy conditions. The methodology of Remer et al. (2012) required only that the specific 1-km pixels making up a given measurement footprint were cloud-free. In practice, if there is a single gap in the clouds of exactly 7 km x 7 km, it is highly unlikely that TROPOMI would be able to get a successful retrieval. Yes, it is officially "cloud-free", but this is treating clouds like a 2D mask, when in reality they are 3-dimensional, with multiple layers, and the sun is very rarely exactly at nadir, in fact never for this domain, and the geostationary imagers are likewise observing at an angle. Thus the light path requires a larger cloud-free area than the ground footprint would suggest. Most retrieval teams find that it is difficult to get good retrievals from very small gaps in clouds due to these problems as

well as light path effects related to nearby clouds and cloud shadows that decrease the amount of reflected light from the ground, reducing the signal to noise. Perhaps some of these effects are less critical for aerosol retrievals, the focus of Remer et al. (2012), but for highly exacting retrievals of greenhouse gases they can be critical.

In addition to this, the footprints given are all at nadir, and in fact they may be somewhat larger depending on the viewing angle. This simple geometry requires a larger gap in the clouds than the footprint alone suggests. It seems the numbers used in this study are taken from Figure 6 in Remer et al., (2012); Figure 7 of the same paper addresses the off-nadir difference for spring, which results in a reduction of 4% for 4 km x 4 km footprints (from 0.31 to 0.27 for MAM, a relative decrease of 13% ).

Thus the effective gap size needed for a 3 km x 3 km instrument may well end up being closer to 8 km x 8 km. This inflation of the footprint size is particularly important for single measurments in broken cloud conditions - this extra padding has its greatest impact around the edge of a cloud-free area. This suggest that the number of cloud-free soundings is likely overestimated. The fact that the median number of observations per model pixel is only 3 for TROPOMI suggests that even a slight reduction in acceptable pixels might have very serious effects on the information content for this instrument. The greater "oversampling" relative to the model resolution for the other instruments means that this will likely have a less serious effect. This is consistent with the information in Figure 4.

One final facet to this discussion that may be worth mentioning is the fact that the actual footprint of TROPOMI may well be 3.5 km x 7 km in the end, due to sampling changes to deal with saturation of the optical channels, which only became apparent post-launch. While this reduction in footprint also results in a smaller signal to noise (and presumably a larger measurement uncertainty), the conclusions on P5, L5-7 suggest that this will likely not degrade the result significantly. I would not redo any of the analysis based upon this information, but simply mention it in the discussion.

Minor comment:

Figure 2: The inclusion of the temporal correlation of cloud cover map in the temporal correlation of modelling error is unnecessarily confusing. Both plots are relevant and worth including, but they should be separated.

Figure 3: The three GeoCARB points should be labelled with x1/day, x2/day, x4/day. The information is contained on the y-axis, but adding this information would make the figure easier to interpret.

Figure 3&4: replot with harmonized colours, so that the same colour always represents e.g. GEO-CAPE, TROPOMI, GeoCARB 1/day, etc.

Watch that the capitalization of GeoCARB is consistent, see e.g. title and legend in Figure 4.

---

## Author Comment (AC1) · 22 Sep 2018

The comment was uploaded in the form of a supplement: https://www.atmos-meas-tech-discuss.net/amt-2018-121/amt-2018-121-AC1-supplement.pdf

---

## Author Response (AR1)

The paper by Sheng et al. examines the information content on methane (CH4) emis- sions contained in column-average concentration measurements by three satellite con- figurations. These configurations reflect the TROPOMI mission in low-Earth-orbit and the GeoCarb and GeoCAPE missions in geostationary orbit. The information content is estimated by a Bayesian inversion for simulated measurements above the Southeast US for a week in summer.

The paper is well written and interesting for the readers of Atmospheric Measurement Techniques in particular since the study can serve as reference for how to size future satellite techniques in terms of spatiotemporal resolution. Therefore, I recommend publishing the paper after considering my comments below:

The paper is a case study for 1 week of CH4 emissions in the Southeast USA. How representative is this case study for the overall challenge of inversely estimating methane emissions on regional scales globally for all seasons? The study would gain scientific mass by extending to other regions and other seasons.
We choose Southeast US and the time period because we took advantage of the previous SEAC4RS study (in order to compare DOFS from satellites with that from SEAC4RS aircraft campaign). We think the region is representative because it accounts for more than 50% US methane emissions with mixed sources from wetlands, oil/gas, coal mines, agriculture, and waste. Summer is when wetlands emissions are highest (anthropogenic emissions are unlikely to have large seasonality). We updated the abstract and text accordingly.

The Bayesian inversion essentially is controlled by the weighting between the measurement uncertainty and the a priori uncertainty. While the assumed measurement uncertainties are described in quite some detail, the text is sparse for the a priori un- certainties. I recommend elaborating in more detail how large the assumed a priori uncertainties are, e.g. a map would help. Is the uncertainty relative to the a priori fluxes i.e. vanishing a priori fluxes remain zero? Likewise, it would be helpful to illustrate the effect of Gaussian Mixture Model used for spatial binning. This information should be included even if it is redundant with previous publications.
We updated Fig. 1 showing prior uncertainties, and we also expanded the discussion on the Gaussian mixture model, and added a figure showing the state vector elements.

The performance analysis focusses on the DOF which is a very condensed measure. I would recommend extending the analysis to the a posteriori flux errors (or the error reductions wrt. the a priori). Could it be enlightening to plot the averaging kernel matrix for cloudy and less cloudy conditions to illustrate the effects of clouds on the information content?
The concept of DOFS is analogue to relative reductions in error variances for the state vector elements. We now add this in the text, and also add a figure showing average kernels (diagonal elements) under zero-cloudy vs cloudy conditions.

The assumed ground-pixel sizes (table 1) are valid for the sub-satellite point (to the best of my knowledge). For wide-swath LEO missions such as TROPOMI, ground- pixels grow substantially

toward the outer parts of the swath. Likewise for GEO, ground- pixel sizes grow with latitude and longitude away from the sub-satellite point. In that sense, the study is too optimistic with respect to the real satellite performance (cloud contamination, measurement density).

We now mention this limitation in the text.

Figure 2, right panel: The inset is somewhat misleading since intuitively one would expect the inset to show kind of the same quantity as the main figure. But, in fact, it is TCCON-model departures in the main figure and cloud cover in the inset. It took me a while to get it. Consider making it separate figures.

We separate the figures now.

The most recent publication for TROPOMI CH4 (real data) is Hu et al., https://doi.org/10.1002/2018GL077259, 2018

We now add Hu et al. (2018).

**J. Marshall (Referee #2)**

This paper presents in a very compact nature a methodology for comparing three differ- ent satellite missions working to constrain methane fluxes using Degrees of Freedom for Signal (DOFS) as a metric of the resolvable information content. This methodology is applied in OSSEs looking at the relative performance of TROPOMI, GeoCARB, and GEO-CAPE. The approach is interesting, and provides a slightly different assessment than the usual reduction of posterior uncertainty. Nonethesless, I have a few concerns regarding some of the assumptions made (particularly with respect to the influence of cloud cover on measurement yield) and the presentation of the results. While the writ- ing is quite clear and free from errors, some additional information is required to help the reader truly understand the approach. (Perhaps the manuscript is a bit too com- pact?) Even after reading some of the referenced papers in a search for explanation, the interpretation of the results was somewhat difficult. As such, some additional infor- mation is requested, as outlined below. If these points are addressed, I would consider the paper suitable for publication in AMT.

In particular, some physical interpretation of the state vector elements developed using the Gaussian Mixture Method (GMM) with Radial Basis Functions (RBFs) would be helpful. The paper in which this method was developed (Turner and Jacob, 2015) is mathematically rather dense, but does provide some information about what these functions look like for California. Having some idea about the relevant processes and the spatial distributions that might be resolved in the study domain used here would be useful. Was temporal aggregation performed as well, over the week, or was a stationary solution assumed? How would this methodology be extended to different time periods or regions? Would the state vector have significantly more or less elements for other similarly-sized domains?

We only focus on spatial aggregation and assume the state vector has no temporal dimension. We added a new figure and expanded the discussion in the text (also see response to review #1)

P3, L7-9: I disagree with the statement that the assumption of randomness in the noise of synthetic measurements does not affect a comparative analysis of different instru- ments. This is true if the instruments which are being compared are expected to have similarly correlated or uncorrelated errors in their actual measurements, but this may well not be the case. An example of this is active vs. passive sensors, where the former is expected to have considerably less correlation between individual measure- ments. While such an assumption has often been made in the past, more experience with satellite measurements have proven time and again that systematic (correlated) errors are incredibly important when trying to interpret signals, and they are not identical across instruments. Please discuss explicitly the limitations of this assumption.
We removed the statement and mentioned this limitation in the text.

P4, L3-10: In the discussion of the model transport error variance, the approach seems valid, but I wonder about its broader applicability over the full domain. Other studies (using in situ data) have shown that the uncertainty dominated by transport errors tends to be proportional to the mean mixing ratios for a given period (see e.g. Jeong et al., JGR, 2013). How representative are the signals at Lamont for the whole domain?
We think it's representative because it's consistent with other studies using real GOSAT data in different regions (California and the North America). We updated the text accordingly.

P4, L11-17: In this discussion of the temporal correlation of the measurements, please explain the increase at around 12 hours seen in the second panel of Figure 2. Does this have something to with the fact that the TCCON measurements are made only during the day, and as such there are fewer samples at around 12 hours? Or is this the result of the well-known smily/frowny tendencies of some TCCON sites at high solar zenith angles? Or an airmass dependency that is not properly accounted for when comparing the modelled fields to the TCCON data? This peak in correlation at 12 hours could be the result of neglecting to apply the (solar-zenith-angle-dependent) averaging kernels to the modelled fields before performing the comparison. Was this done? Some more detail is needed here.
This increase may be due to several reasons as mentioned above, but it's not our main interest. We used an exponential fit (largely driven by the first 12 hours) to compute the correlation time scale, and this increase around 12 hour has little impact on our results. We now explain this in the text.

P4, L22-P5, L7: The discussion of the cloud cover is probably the most critical point in this manuscript, upon which many of the conclusions rest. I expect that the estimation of number of successful retrievals is overestimated for partially cloudy conditions. The methodology of Remer et al. (2012) required only that the specific 1-km pixels making up a given measurement footprint were cloud-free. In practice, if there is a single gap in the clouds of exactly 7 km x 7 km, it is highly unlikely that TROPOMI would be able to get a successful retrieval. Yes, it is officially "cloud-free", but this is treating clouds like a 2D mask, when in reality they are 3-dimensional, with multiple layers, and the sun is very rarely exactly at nadir, in fact never for this domain, and the geostationary imagers are likewise observing at an angle. Thus the light path requires a larger

cloud- free area than the ground footprint would suggest. Most retrieval teams find that it is difficult to get good retrievals from very small gaps in clouds due to these problems as well as light path effects related to nearby clouds and cloud shadows that decrease the amount of reflected light from the ground, reducing the signal to noise. Perhaps some of these effects are less critical for aerosol retrievals, the focus of Remer et al. (2012), but for highly exacting retrievals of greenhouse gases they can be critical.

We now acknowledge this limitation in the text, but we don't think they can be critical for our results. Actually we did a sensitive test using different cloudy conditions (cloud fraction from 0.5 to 0.8). As we shown in Fig. 6, geostationary instruments are insensitive to cloudy conditions on the regional scale (~25km), though TROPOMI is sensitive. Our retrieval rate for synthetic TROPOMI (7x7 km2) is similar to that of GOSAT full-physic retrievals. We now explicitly discuss this in the text.

In addition to this, the footprints given are all at nadir, and in fact they may be somewhat larger depending on the viewing angle. This simple geometry requires a larger gap in the clouds than the footprint alone suggests. It seems the numbers used in this study are taken from Figure 6 in Remer et al., (2012); Figure 7 of the same paper addresses the off-nadir difference for spring, which results in a reduction of 4% for 4 km x 4 km footprints (from 0.31 to 0.27 for MAM, a relative decrease of 13% ). Thus the effective gap size needed for a 3 km x 3 km instrument may well end up being closer to 8 km x 8 km. This inflation of the footprint size is particularly important for single measurments in broken cloud conditions - this extra padding has its greatest impact around the edge of a cloud-free area. This suggest that the number of cloud-free soundings is likely overestimated. The fact that the median number of observations per model pixel is only 3 for TROPOMI suggests that even a slight reduction in acceptable pixels might have very serious effects on the information content for this instrument. The greater "oversampling" relative to the model resolution for the other instruments means that this will likely have a less serious effect. This is consistent with the information in Figure 4.

We now mention this limitation of footprint size in the text (also see response to Review #1).

One final facet to this discussion that may be worth mentioning is the fact that the actual footprint of TROPOMI may well be 3.5 km x 7 km in the end, due to sampling changes to deal with saturation of the optical channels, which only became apparent post-launch. While this reduction in footprint also results in a smaller signal to noise (and presumably a larger measurement uncertainty), the conclusions on P5, L5-7 suggest that this will likely not degrade the result significantly. I would not redo any of the analysis based upon this information, but simply mention it in the discussion.

The footprint size for CH4 might be further reduced to ~5.5km x 7 km in the final product (Ilse Aben, personal communication, 2018). We now mention it in the text.

Minor comment:

Figure 2: The inclusion of the temporal correlation of cloud cover map in the temporal correlation of modelling error is unnecessarily confusing. Both plots are relevant and worth including, but they should be separated.

We have separated the figure.

Figure 3: The three GeoCARB points should be labelled with x1/day, x2/day, x4/day. The information is contained on the y-axis, but adding this information would make the figure easier to interpret.
We updated the figure.

Figure 3&4: replot with harmonized colours, so that the same colour always represents e.g. GEO-CAPE, TROPOMI, GeoCARB 1/day, etc.
We updated the figures.

Watch that the capitalization of GeoCARB is consistent, see e.g. title and legend in Figure 4
We corrected it.

[revised manuscript text omitted]

---

## Author Response (AR2)

Dear Dominik,

Thank you for your comments.

The pixel resolution in the original GeoCarb design was 2.7 km (north-south) x 6 km (east-west), but the EW spacing between scans was only 3 km so the actual resolution was 2.7x3.0 km2. This is stated in Section 3.1 of O'Brien et al. (2016): "… When viewing to nadir, the east-west scan step is 3.0 km, and the north-south spacing of pixels is 2.7 km…". Therefore, we think our use of 3x3 km2 is fine.

[revised manuscript text omitted]

---

## Author Response (AR3)

Dear Dominik,

Thank you for your comment.

[revised manuscript text omitted]